# Different Drought Legacies of Rain-Fed and Irrigated Croplands in a Typical Russian Agricultural Region

**Yuanhuizi He [1,2], Fang Chen [2,3,*], Huicong Jia [2,3], Lei Wang [1] and Valery G. Bondur [4]**

1   State Key Laboratory of Remote Sensing Science, Aerospace Information Research Institute, Chinese Academy of Sciences, Beijing 100094, China; heyhz@radi.ac.cn (Y.H.); wanglei@aircas.ac.cn (L.W.)
2   University of Chinese Academy of Sciences, Beijing 100049, China; jiahc@radi.ac.cn
3   Key Laboratory of Digital Earth Science, Aerospace Information Research Institute, Chinese Academy of Sciences, Beijing 100094, China
4   Institute for Scientific Research of Aerospace Monitoring "AEROCOSMOS", Gorokhovsky Pereulok 4, 105064 Moscow, Russia; vgbondur@aerocosmos.info
*   Correspondence: chenfang_group@radi.ac.cn; Tel.: +86-10-8217-8105

**Abstract:** Droughts are one of the primary natural disasters that affect agricultural economies, as well as the fire hazards of territories. Monitoring and researching droughts is of great importance for agricultural disaster prevention and reduction. The research significance of investigating the hysteresis of agricultural to meteorological droughts is to provide an important reference for agricultural drought monitoring and early warnings. Remote sensing drought monitoring indices can be employed for rapid and accurate drought monitoring at regional scales. In this paper, the Moderate Resolution Imaging Spectroradiometer (MODIS) vegetation indices and the surface temperature product are used as the data sources. Calculating the temperature vegetation drought index (TVDI) and constructing a comprehensive drought disaster index (CDDI) based on the crop growth period allowed drought conditions and spatiotemporal evolution patterns in the Volgograd region in 2010 and 2012 to be effectively monitored. The causes of the drought were then analyzed based on the sensitivity of a drought to meteorological factors in rain-fed and irrigated lands. Finally, the lag time of agricultural to meteorological droughts and the hysteresis in different growth periods were analyzed using statistical analyses. The research shows that (1) the main drought patterns in 2010 were spring droughts from April to May and summer droughts from June to August, and the primary drought patterns in 2012 were spring droughts from April to June, with an affected area that reached 3.33% during the growth period; (2) local drought conditions are dominated by the average surface temperature factor. Rain-fed lands are sensitive to the temperature and are therefore prone to summer droughts. Irrigated lands are more sensitive to water shortages in the spring and less sensitive to extremely high temperature conditions; (3) there is a certain lag between meteorological and agricultural droughts during the different growth stages. The strongest lag relationship was found in the planting stage and the weakest one was found in the dormancy stage. Therefore, the meteorological drought index in the growth period has a better predictive ability for agricultural droughts during the appropriately selected growth stages.

**Keywords:** temperature vegetation drought index (TVDI); comprehensive drought disaster index (CDDI); crop drought; meteorological droughts; spatiotemporal evolution

## 1. Introduction

A drought is defined as a prolonged and abnormal moisture deficiency [1]. This recurring phenomenon has a variety of geographical and temporal distributions, which have an impact on

the natural environment, ecosystem and economic production, and life [2]. With the occurrence of global warming over the past 100 years, climate change has exacerbated the occurrence of extreme events, such as droughts. Droughts have developed a wider extension, longer durations, and greater severities [3]. In the past few decades, the frequency and extension of regional droughts have increased [4]. The current state of drought occurrences has caused drought-related research to become an important issue in academia [5]. Droughts can be divided into four categories based on their impacts and characteristics, which are denoted as meteorological droughts, agricultural droughts, hydrological droughts, and socioeconomic droughts [6,7]. These four types of droughts are complicated and interrelated phenomena, which propagate in different ways with varied definitions [5]. A meteorological drought is often defined from the perspective of the degree and duration of the lack of precipitation [6]. It focuses on the reduction of surface water supply directly caused by precipitation deficit, as well as the water deficit caused by an abnormal temperature, which intensify surface evapotranspiration. Short-term (i.e., a few weeks duration) meteorological dryness is the key trigger that depletes soil moisture storage and suppresses root water absorption, and is thus closely linked to the onset of an agricultural drought [2]. However, an agricultural drought is far more specialized and complicated owing to its dependence on prevailing meteorological conditions, biophysical characteristics, growth stages, and other factors, such as soil properties [6]. Definitions of a hydrological drought are concerned with the effects of dry spells on the surface or sub-surface hydrology [6]. They focus on evaluating the period during which the groundwater level and surface runoff are inadequate for maintaining supply for an established use under a given water management system [8]. Meteorological dryness is usually the starting point of hydrothermal anomalies. In general, a prolonged precipitation insufficiency generates less input for the hydrological system [7]. Distinct from the other three types of drought, a socioeconomic drought focuses on the impacts of drought events on the social, economic, and ecological environment, which are more complicated and difficult to quantify due to the impact of social production and human activities. These four drought types are interactive in the water cycle [9] and related to different durations of the lack/reduction of precipitation and the impacts that are progressively caused. An agricultural drought with regards to crops is an important natural disaster that threatens food security and sustainable agricultural development due to the lack of precipitation or irrigation, insufficient soil moisture, and other factors in crop growth [10]. As a result, two drought types are often considered together to assess the spatiotemporal evolution and development model of drought events in a given region.

Traditional drought monitoring is usually denoted as meteorological drought monitoring, i.e., constructing drought indexes, such as the precipitation anomaly percentage (Pa) [11], the Palmer drought severity index (PDSI) [12], the standardized precipitation index (SPI) [13], and the standardized precipitation evapotranspiration index (SPEI) [14], over several years, based on meteorological and hydrological data from ground observation sites or other precipitation data. This method allows accurate reflections of meteorological changes on a multi-year scale based on meteorological stations. However, the occurrence of droughts has obvious spatiotemporal characteristics, and it is difficult to accurately estimate drought events on a large spatial scale using limited site data and spatial interpolation methods with bias. Remote sensing (RS) ground observations using relevant image processing technology [15,16] can provide accurate estimates for natural disasters [17,18], such as drought events on large spatiotemporal scales, based on multi-source data and multiple drought indices. The principle of the remote sensing drought index is to detect and indicate ground targets from satellite data, such as the land surface temperature, soil moisture, crop physiological parameters, and cloud cover, in order to generate the corresponding index models and monitor the long-term spatiotemporal evolution dynamics of droughts. Various types of indices are used to analyze the state of soil and vegetation according to remote sensing data. This RS approach, which is based on the use of brightness values in various spectral channels of satellite instruments, is capable of producing dryness information and an auxiliary risk assessment for arid areas and the ensuing potential disasters, such as forest fires [19–23].

Different types of remote sensing drought indices exist. They can be grouped by the type of drought parameters considered (precipitation, moisture, and evapotranspiration), the impacts of a drought on vegetation (abiotic stresses), or the approach (energy balance, water balance, etc.). They consist of the following: (1) Describe soil moisture changes: Apparent thermal inertia (ATI) [24] and the(Modified) perpendicular drought index (PDI/MPDI) [25–27]; (2) describe canopy temperature changes and energy balance: The temperature condition index (TCI) [28], crop water stress index (CWSI) [29], water deficit index (WDI) [30], temperature vegetation dryness index (TVDI) [31], modified temperature vegetation dryness index (MTVDI) [32,33], drought severity index (SDI) [34], vegetation supply water index (VSWI) [35], etc.; (3) describe changes in the crop morphology and greenness: The normalized difference vegetation index (NDVI) [36], vegetation condition index (VCI) [37], standard vegetation index (SVI) [38], anomaly vegetation index (AVI) [39], Enhanced Vegetation Index (EVI) [40], etc.; (4) describe vegetation moisture content changes based on the shortwave infrared band (SWIR): The shortwave infrared perpendicular water stress index (SPSI) [41] and global vegetation moisture index (GVMI) [42]; (5) describe the moisture content using microwave technology: The microwave integrated drought index (MIDI) [43], microwave polarization index (MPI) [44], and Microwave Temperature Vegetation Drought Index (MTVDI) [45].

Other scholars have combined vegetation, temperature, and precipitation variables in remote sensing data to build comprehensive drought indices, such as the combined deficit index (CDI) [46–48], scaled drought condition index (SDCI) [49], synthesized drought index (SDI) [50], and optimized vegetation drought index (OMDI and OVDI) [51], which are relatively consistent with the meteorological station data. A normalized difference water index (NDWI) [52] time series can be constructed to derive the standardized water index (SWI) [53] or be combined with the NDVI for drought monitoring [54]. Among these indices, the TVDI is a remote sensing drought index based on a combination of the vegetation index and temperature and has clear physical and biological significance, with parameters that are easier to obtain. This is suitable for surface drought monitoring with large changes in the vegetation coverage and is well-correlated with the soil moisture [55]; thus, it has been widely applied in actual drought monitoring [56–59].

Russia is an important agricultural country located in the "Belt and Road" economic zone, with high drought levels each year [60], and is affected by various natural and social factors, such as climate, terrain, and irrigation. Extreme dry weather from 2010 to 2012 [61] had a significant impact on the crop yields [62], the intensity of wildfires [63–67], and emissions of harmful gases and aerosols into the atmosphere caused by these wildfires [68,69], as well as the agricultural economy in important food-producing areas of Russia [70], especially in the Volga and Don River basins of Volgograd [71,72], which is the primary grain-producing region in the southern part of the country. Therefore, monitoring agricultural droughts in the Volgograd region can effectively help perform scientific drought assessments, which is important for the stability of the agricultural economy and ensuring the sustainable development of agricultural production.

The main research objectives of this work are as follows: (1) To evaluate the ability of the TVDI and the comprehensive drought disaster index (CDDI) to identify drought conditions in the Volgograd region from 2010 and 2012; (2) to analyze the drought evolution pattern of rain-fed and irrigated lands combined with climatic factors and the meteorological drought index; and (3) to analyze the hysteresis of agricultural to meteorological droughts and the lag time for each growth period.

## 2. Materials and Methods

### 2.1. Study Area

The Volgograd region is located in the southeastern part of the Eastern European Plain and has an area of 113,900 square kilometers at the lower reaches of the Volga River Basin in Russia. The primary rivers in the state are the Volga and Don Rivers, and the Volga-Don canal. The region has a hot summer continental climate, with a warm to hot summer and cold winter. The precipitation shows

seasonal changes of concentrated rain in summer, little rain in spring and autumn, and snow cover in winter. The annual precipitation decreases from the northwest to the southeast, with an average yearly precipitation of 348 mm. There is a large range of temperatures, from the highest record of 42.6 °C to the lowest record of −33.0 °C, with an average of 8.2 °C [73]. The arable land in this area accounts for approximately half of the land and is one of Russia's main crop-producing and high-yield regions. The important crops include wheat, barley, and corn, and are widely dispersed in the region. Based on the United States Geological Survey's 1-km global map of major crops [74], the northern area of the river primarily exhibits rain-fed agriculture dominated by wheat, and the southern area of the river mainly includes irrigated agriculture. Planted corn crops and other vegetation types, such as grasslands, are distributed in a small area to the east. The geographical location of the area and the distribution of farmland types are illustrated in Figure 1.

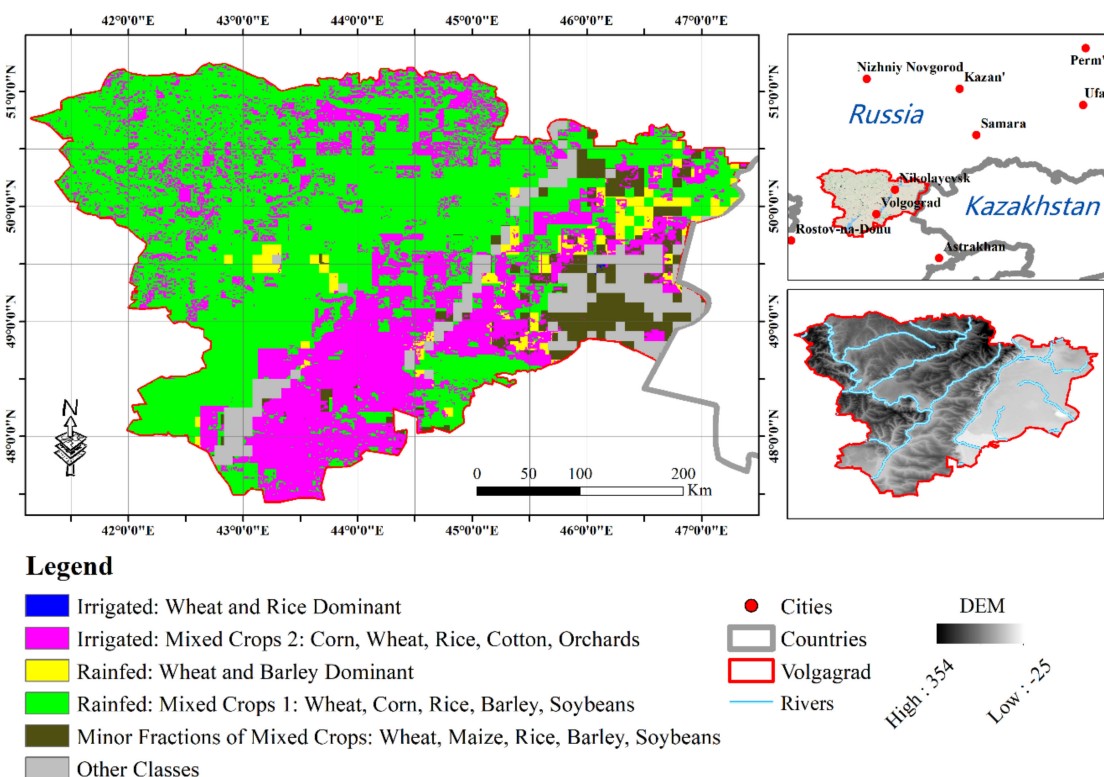

**Figure 1.** Location and farmland distribution of the Volgograd region.

Figure 2 displays a phenology calendar of the main crops in the area [75]. The growing cycle of crops can be roughly divided into three stages, except for dormancy, which is hardly related to crop droughts: (1) Planting stage from April to May: The main crops in the region enter a period of planting around April with an increased temperature. This is a vegetative growth stage dominated by root development and leaf emergence, during which the crop water requirement mainly comes from the shallow soil moisture absorbed by the root system; (2) flowering stage from June to early August: Abundant solar radiation and plentiful rainfall greatly enhance the photosynthesis process, which brings plants into a key period of the whole growth cycle, including heading, flowering, and filling stages. The increase of the soil surface temperature enhances soil evaporation, while the transpiration of vegetation increases the consumption of soil water content. Therefore, crops such as winter wheat and corn are the most sensitive to temperature and moisture stress during this period [76]; (3) harvest stage from late July through the end of October: After entering autumn in September, the main crops gradually enter the harvest period, with fewer demands for moisture and temperature.

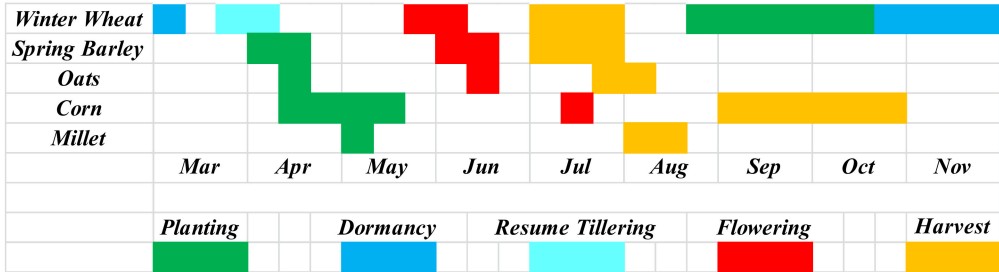

**Figure 2.** Major crops and phenology calendar of the Volgograd region.

## 2.2. Satellite Imagery and Auxiliary Data

The remote sensing data utilized in this article come from the National Aeronautics and Space Administration (NASA) and includes the MODIS13A2 product of the NDVI and the MODIS11A2 product of the land surface temperature for the Terra satellite from 2008–2012, with a spatial resolution of 1 km × 1 km. The 0.25° × 0.25° PERSIANN-CDR [77,78] precipitation data from this area were used for auxiliary analyses of the drought conditions. The Google Earth Engine (GEE) platform was applied to generate the MODIS13A2 normalized vegetation index and the MODIS11A2 land surface temperature using scaling factors. The image data were then downloaded from vector boundary maps of the study area. To understand the impacts of droughts on different farming areas and the different crop types, this paper also refers to the auxiliary land cover classification data to enable stratification analyses, which include the Global Food Security-Support Analysis Data at 1 km × 1 km (GFSAD30) [74] released by the United States Geological Survey and the Remote Sensing Mapping Data Products for Global Coverage of 30 m × 30 m with a base year of 2010 [79–81], published by the Basic Geographic Information Center. Details of the remote sensing data and auxiliary data are shown in Tables 1 and 2, respectively.

**Table 1.** Details of the remote sensing data.

| Data Set | Coverage | Period | Frequency | Resolution |
|----------|----------|--------|-----------|------------|
| MOD13A2 | Global | 2000–present | 16-daily | 1 km × 1 km |
| MOD11A2 | Global | 2000–present | 8-daily | 1 km × 1 km |
| PERSIANN-CDR | 60°S–60°N | 1983–present | Daily | 0.25° × 0.25° |

**Table 2.** Details of the auxiliary data.

| Auxiliary Data Set | Coverage | Period | Categories | Resolution |
|--------------------|----------|--------|-----------|------------|
| GFSAD30 | Cropland extent | 1990–2017 | Five dominant crop types | 1 km × 1 km |
| GlobeLand30-2010 | 80°S–80°N | 2010 | 10 types of surface cover | 30 m × 30 m |

## 2.3. Temperature Vegetation Drought Index (TVDI)

The TVDI [31] primarily applies to the NDVI-LST space (a two-dimensional spatial scatterplot of the NDVI and LST values of all the pixels in a region) by combining the temperature and vegetation index products for remote sensing data when the vegetation coverage and soil moisture conditions in the study area change significantly. Monitoring the drought effects for the TVDI index in different regions is often accomplished by fitting the wet- and dry-edge equations in the feature space, classifying drought conditions, and verifying the authenticity with actual precipitation or soil moisture data. A description and definition of the TVDI were first provided based on the rules presented from the NDVI and LST on a two-dimensional spatial scatter plot, which can be expressed by [31]

$$\text{TVDI} = \frac{\text{LST} - \text{LST}_{\min}}{\text{f(VI)}_{\max} - \text{LST}_{\min}}, \tag{1}$$

$$f(VI)_{max} = a_{max} - b_{max}*VI, \tag{2}$$

where $f(VI)_{max}$ is the dry-edge linear fitting equation of the LST/VI scattered triangle space, $a_{max}$ and $b_{max}$ are the fitting parameters, VI is the vegetation index and the abscissa axis, and LST is the surface temperature and the ordinate axis. The numerator in Equation (1) represents the temperature difference between the actual temperature and the minimum temperature of a pixel under the vegetation coverage, which is represented as A. The denominator represents the temperature difference between the maximum possible temperature and the minimum temperature of a pixel under a certain vegetation coverage, which is represented as B. The geometric meaning of TVDI in the LST/VI scattered point triangle space is shown in Figure 3. Therefore, the TVDI can represent the relatively dry and wet conditions for the pixels, in order to identify and monitor the soil moisture and drought conditions.

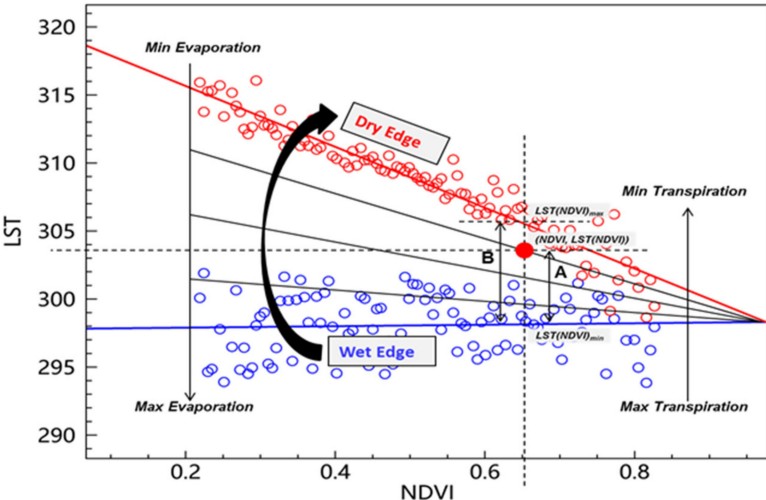

**Figure 3.** The temperature vegetation drought index (TVDI) definition and the geometric meaning at a given pixel.

The TVDI value varies from 0 to 1. When TVDI = 0, the pixel falls on the wet edge of the soil line. In this case, the surface humidity is greater, with the strongest evaporation and vegetation transpiration, and the soil is less affected by drought. When TVDI = 1, the pixel is on the dry edge of the soil line with the lowest surface humidity, the weakest effects of evaporation and vegetation transpiration, and a higher degree of soil drought. The TVDI monitoring results are classified as shown in Table 3, based on the MODIS data temperature vegetation drought index classification [82].

**Table 3.** Different levels of the TVDI drought index.

| TVDI | Levels | Soil Moisture Status |
|---|---|---|
| $0 < TVDI < 0.46$ | No drought | Surface water is sufficient or normal |
| $0.46 \leq TVDI < 0.57$ | Mild drought | Small amount of surface evaporation and dry air near the surface |
| $0.57 \leq TVDI < 0.76$ | Moderate drought | Soil surface is dry and vegetation leaves are wilting |
| $0.76 \leq TVDI < 0.86$ | Severe drought | Thicker dry soil layers appear, and vegetation is withered |
| $0.86 \leq TVDI < 1$ | Extreme drought | Surface vegetation is dry or dead |

*2.4. Percentage of Precipitation Anomaly (Pa)*

A decrease in rainfall or increased surface warming within a certain period will cause a meteorological drought, and its accumulation and development will affect the soil moisture content and the crop water demand process. Therefore, a meteorological drought is an important factor that directly triggers an agricultural drought. These two types of droughts and their lag correlation [83–86] are often comprehensively considered to evaluate the spatial and temporal evolution and development of drought events in a region. This paper applies the Pa to indicate the development process of a

meteorological drought. The lag of the agricultural drought is indicated by the TVDI index and the occurrence of drought events is compared with the Pa. The formula employed to calculate the Pa in a certain period is

$$\text{Pa} = \left[ \left( P - \overline{P} \right) / \overline{P} \right] * 100\%, \tag{3}$$

where P is the precipitation over a certain period and $\overline{P}$ is the average precipitation in that same period. Pa can represent a precipitation deviation from the average level in a specific location and can concisely represent the meteorological drought caused by abnormal decreases in precipitation.

*2.5. Comprehensive Drought Disaster Index (CDDI)*

The occurrence of a regional drought is affected by the cumulative drought caused by hydrothermal conditions during a particular period. It is also a comprehensive response to the sensitivity of vegetation and crops in the region to water scarcity in different growing seasons. These two aspects of drought sensitivity at key growth stages can provide a CDDI that depends on the crop growth stages. According to the sensitivity of crops to rainfall at different growth stages, this index sets the weight of the drought situation at each stage, and then calculates the final drought situation index by weighted summation. Based on the research of related scholars on the sensitivity of crops to the lack of rain at key growth stages [87] and the phenology calendar information for the primary crops in the study area, the main stages of crop growth occur from April to August. This period is also when the crop responds most strongly to hydrothermal climate conditions. The TVDI during the growth period of April to August is used to set the weight value for each month to indicate the sensitivity of crops to the drought response during different growth periods. The weighted calculations allow a CDDI to be constructed to evaluate the regional drought situation. The construction calculation formula is

$$\text{CDDI} = \sum_{I=4}^{8} P_i * \text{TVDI}, \tag{4}$$

where $P_i$ is the weight value of each month. Combined with the existing crop drought sensitivity settings [87] and the main crop phenology calendar in the study area, the weights $P_i$ from April to August were set to 0.4, 0.5, 0.8, 0.9, and 0.4, respectively. The drought index classification basis was used to obtain the drought classification via linear weighting, as displayed in Table 4.

**Table 4.** Different levels of the comprehensive drought disaster index (CDDI).

| April to May | May to June | April to August | Levels of Drought Disaster |
|---|---|---|---|
| $0 < \text{CDDI} < 0.828$ | $0 < \text{CDDI} < 1.932$ | $0 < \text{CDDI} < 2.76$ | No drought |
| $0.828 \le \text{CDDI} < 1.026$ | $1.932 \le \text{CDDI} < 2.394$ | $2.76 \le \text{CDDI} < 3.42$ | Mild drought |
| $1.026 \le \text{CDDI} < 1.368$ | $2.394 \le \text{CDDI} < 3.192$ | $3.42 \le \text{CDDI} < 4.56$ | Moderate drought |
| $1.368 \le \text{CDDI} < 1.548$ | $3.192 \le \text{CDDI} < 3.612$ | $4.56 \le \text{CDDI} < 5.16$ | Severe drought |
| $1.548 \le \text{CDDI} < 1.8$ | $3.612 \le \text{CDDI} < 4.2$ | $5.16 \le \text{CDDI} < 6$ | Extreme drought |

## 3. Results

*3.1. Time Evolution of TVDI Drought Monitoring*

A crop drought is a concentrated reflection of the local hydrothermal conditions over a particular period. The comparison of hydrothermal conditions indicated in Figures 4 and 5 over the same period in different years indicates that the precipitation in 2010 was lower than in the same period of the other years, especially from June to August. At the same time, the surface temperature data from July to September of 2010 also showed higher temperature levels. The precipitation was deficient from April to June 2012, and the temperature for the entire crop growing season was relatively high. These higher temperatures and lower rainfall environments increase the probability of crop drought events and the fire hazard of territories [69]. The soil line fitting results in the LST/NDVI feature space provide the TVDI drought monitoring results for each phase of the primary crop growing season from April to

October. The classified TVDI standard was used to generate the time evolution for the retrospective drought events in 2010 and 2012 shown in Figure 6, in order to reflect the actual drought conditions of the surface.

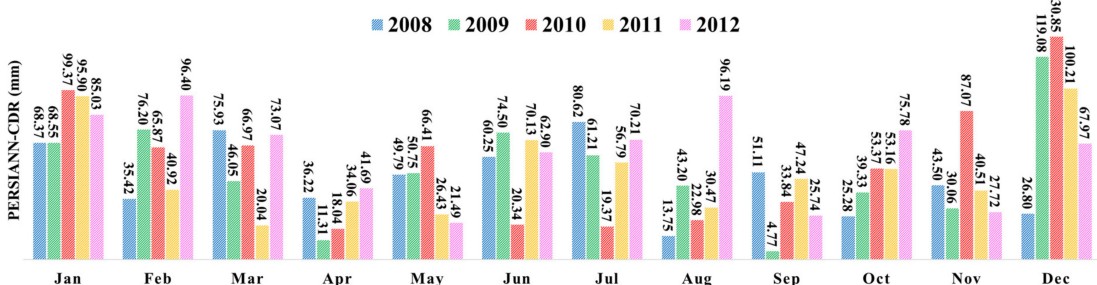

**Figure 4.** Bar chart of monthly cumulative precipitation over the same period from 2008 to 2012.

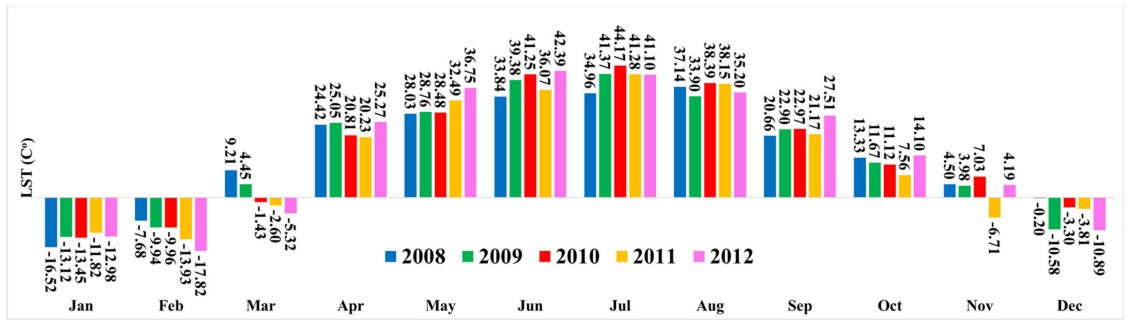

**Figure 5.** Bar chart of the monthly average temperature (LST) over the same periods from 2008 to 2012.

The results of droughts visualized in Figure 6 are closely related to the local conditions. Drought conditions first appeared in spring during April, and there were higher levels of drought from the summer to late September in different croplands and lower levels of drought at the end of the year. Detailed information on the climatic conditions and drought propagation in the primary growth stages is as follows:

(1) Planting stage from April to May: An extreme lack of rain in early April 2010 led to a reduction of the soil water content, which restrained root water absorption and the water supply for leaf growth. This agrometeorological propagation exerted a considerable influence on the heading process of winter wheat, which is widely distributed in the rain-fed drylands to the north of the river. The same process took place in May 2012, during which a precipitation deficit and rapidly rising temperature caused extensive stress to regional crops;

(2) Flowering stage from June to early August: The cumulative precipitation in June and July 2010 was only about 30% of the level in the same period, accompanied by extremely high temperatures in July. Intense soil evaporation led to the soil moisture restraining the biophysical process of crops during flowering. The relative water content of the leaves rapidly decreased under high temperature stress, which led to an imbalance between the water supply and demand and consequently, spreading drought. Crops also suffered from water deficit when rainfall was not absent in 2012. On account of intense solar radiation and an extremely high temperature in June, the potential atmospheric evaporation was quite strong, leading to vigorous crop transpiration. The water absorbed by the root system from the upper soil was inadequate to compensate for transpiration consumption, thus resulting in water deficit in the flowering stage of corn, dominantly cultivated in southern irrigated land;

(3) Harvest stage from late July through to the end of October: The gradual rain rebound and the arrival of a cool harvest autumn restored the soil moisture and alleviated the summer drought. Crops require different levels of water during their various growth stages, and the water demand

is greatest in the early and middle stages of growth. In general, the TVDI drought monitoring results in Figure 6 show different phenomena for the various months. The drought in 2010 primarily occurred in early April and from June to August, and was affected by the simultaneous impacts of high temperatures and low rainfall. The drought in 2012 mainly occurred from April to June and was mostly affected by the lack of precipitation.

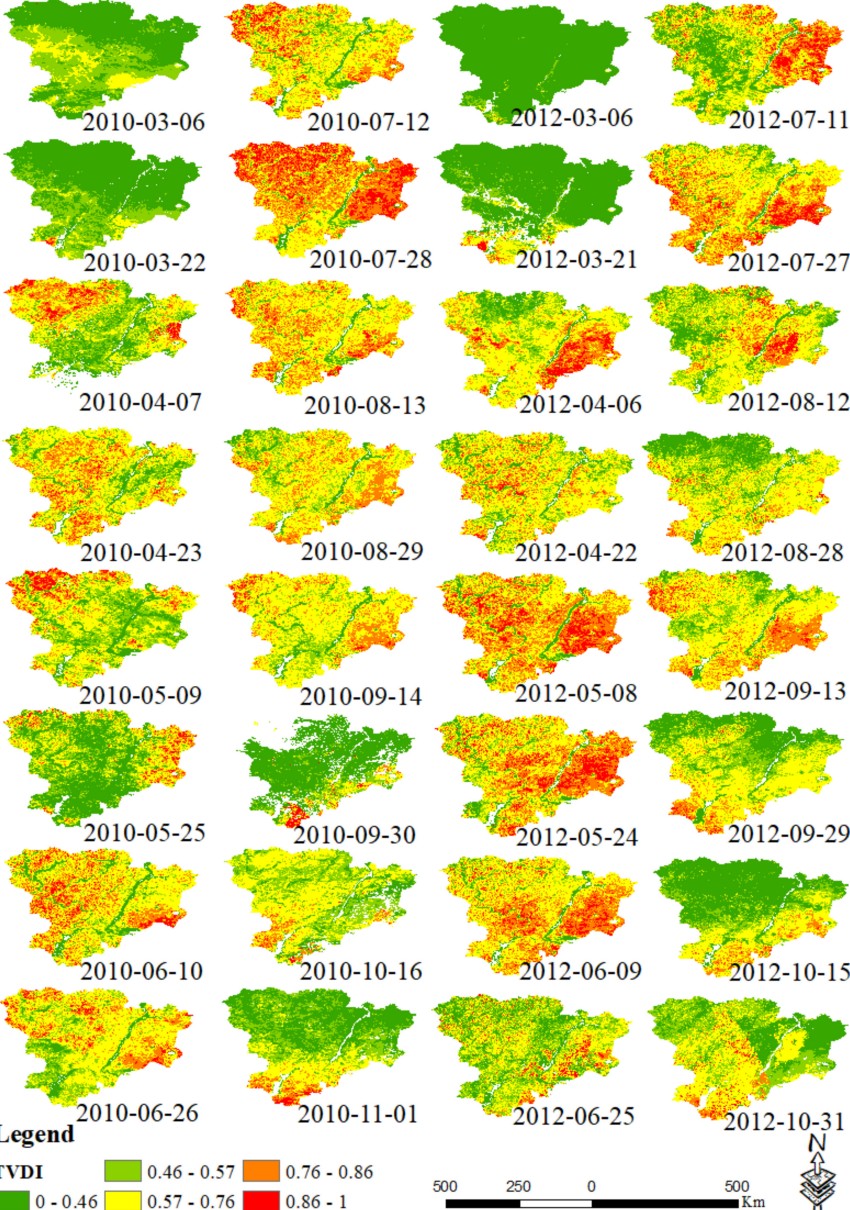

**Figure 6.** TVDI propagation of the major crop growth stages in Volgograd in 2010 and 2012.

*3.2. Factor Analysis of Drought Conditions between Rain-Fed and Irrigated Lands*

The response of the surface to a drought is affected by the local hydrothermal environment and type of surface crops. Rain-fed and irrigated lands are two important types of farming in agricultural production activities. Rain-fed land simply depends on natural rainfall to meet the crop water requirements, and irrigated land considers the intervention of irrigation events due to factors that include insufficient natural hydrothermal conditions. Therefore, the two have different sensitivities to hydrothermal conditions and need to be considered separately when analyzing drought patterns [88].

On the one hand, a TVDI of 0.46 is used as the threshold for drought occurrence in Figure 7. The drought evolution pattern from 2008 to 2012 shows that TVDI changes positively with temperature during the key growth period from April to October. In the middle and late stages of crop growth, the TVDI index showed a significant downward trend with the increase of rain. On the other hand, there are several differences in the drought patterns for 2010 and 2012, which can be described as follows:

(1)   Duration: Irrigated land suffered earlier than rain-fed land in terms of their starting times and durations. Additionally, the degree of drought for the irrigated land was higher than that for the rain-fed land within the considered years, except for 2010. The rain-fed land drought always ended earlier due to its sensitivity to water and heat conditions;

(2)   Climatic factors: The distinction in the drought intensity between the rain-fed and irrigated lands is influenced by disturbances in the average surface temperature and extreme weather events. Figure 7 shows the monthly mean temperature changes for the two types of cultivated lands over the same period. It can be seen that the irrigated land is located in the southeastern region of Volgograd with lower latitudes and is closer to the steppe semi-desert zone in the northern Caspian Sea. Therefore, its average monthly temperature is higher than that of the rain-fed arable land in the northwest for most of the year. Considering this, the local climate sensitivity factors include a higher average surface temperature ($\overline{LST}_{irr} > \overline{LST}_{rain}$), extremely low rainfall (2010 and 2012), and exceptionally high temperatures (2010) [71];

(3)   Sensitivity: Irrigated land is more sensitive to disturbances in the precipitation factor as dominated by the average surface temperature, while the rain-fed land is more sensitive to disturbances in the temperature factor as dominated by the precipitation. This also explains the greater effects of a drought on irrigated land, which appears earlier and lasts longer. Moreover, this also provides a reason for the higher sensitivity of rain-fed land to droughts when extreme high-temperature events occurred in 2010, but the stronger regulation capacity of irrigated land.

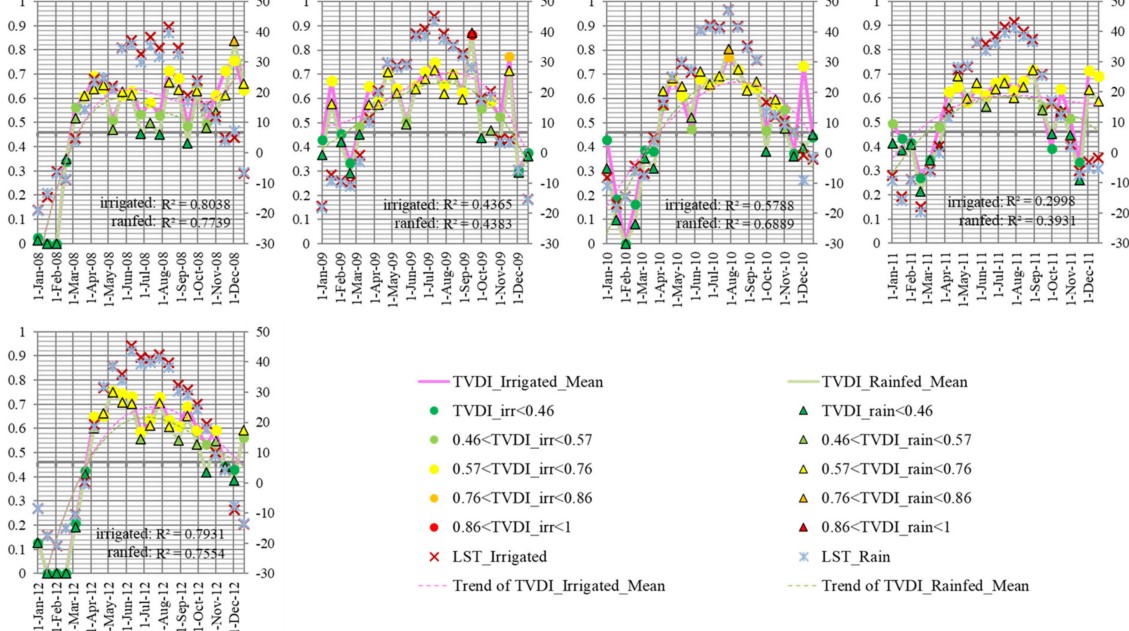

**Figure 7.** Differences in the drought responses of rain-fed and irrigated lands over the years 2008 to 2012.

It can be seen that (1) irrigated land is more prone to spring droughts when conditions are dry with less rainfall. However, the rain-fed land is prone to summer droughts with high temperatures, which is consistent with the previous spatial TVDI distributions. Additionally, (2) the drought conditions

of crops are primarily affected by extreme hydrothermal events, especially temperature [89]. Under the extreme high temperature conditions, the inclusion and regulation of artificial irrigation events in irrigated areas have a greater advantage for suppressing drought conditions.

*3.3. Lagging Relationship between Meteorological and Agricultural Droughts*

On one hand, the key thresholds for meteorological drought and crops affected by drought were analyzed. Figure 7 indicates that, from April to October, the TVDI index was concentrated at levels greater than 0.57, which suggests a moderate drought. With the end of the growing period, the TVDI index also dropped to a level of a mild or no drought. Therefore, a TVDI of 0.57 is the key threshold for crops to experience drought. On the other hand, a Pa of −15% [11] was employed as the threshold to indicate the occurrence of a meteorological drought each year, and a polynomial function was assigned as an approximation based on the data accuracy and actual conditions. Figure 8 shows the drought processes as identified from the Pa and TVDI index each year, as well as the primary phenology periods for local crops [75]. The meteorological drought index Pa is sensitive to changes in the inter-annual precipitation factors, and generally appears in the form of "drying-wetting-drying"; the two drier periods occur at the turn from spring to summer and the turn from autumn to winter. The TVDI index is highly synchronized with the key growth periods (planting and flowering periods) of the crops and is more sensitive to years when crops are affected by droughts.

There is a lag between the occurrence of meteorological droughts and agricultural droughts. Apart from the overall wet year of 2008, meteorological droughts generally began during the decreased rainfall and rising temperature period from March to April and became agricultural droughts for crops after one to two months [48,90]. Figure 8b shows that the meteorological drought in 2009 dropped to −15% on day 73, and the crop drought occurred at the rain-fed (irrigated) land at day 105 (90), which lagged behind the meteorological drought by 32 (17) days. Figure 8c shows that when the Pa in 2010 dropped to the level of −15% on day 80, the entire region exhibited a majority of meteorological drought occurrence. The agricultural drought occurred in the rain-fed and irrigated lands at days 112 and 113, respectively, which were 52 and 53 days behind the meteorological drought. Figure 8d shows that the meteorological drought in 2011 dropped to −15% on day 28, and the agricultural drought occurred at the rain-fed (irrigated) land on day 118 (112), which lagged behind the meteorological drought by 90 (84) days. Figure 8e shows that the meteorological drought in 2012 dropped to −15% at day 77, and the agricultural drought occurred for rain-fed (irrigated) land at day 106 (103), which lagged behind the meteorological drought by 29 (26) days.

3.3.1. Hysteresis at Different Growth Stages between Pa and TVDI

Agricultural and meteorological droughts exhibit hysteresis with characteristics that vary for the different growth periods and types of cultivated land. In this study, the correlation between the meteorological drought index and the agricultural drought index was fit in the four stages of dormancy, planting, flowering, and harvest, as shown in Figure 9. First, the correlation between Pa and TVDI was the lowest during the winter crop dormancy. The time of the agricultural drought represented by the TVDI was relatively stable in this stage, which is from March to April. The critical points appeared in the middle stage of the planting period, which was less dependent on precipitation and more dependent on temperature. Therefore, the lag time for the occurrence of droughts primarily comes from the Pa index critical point. The lag relationship is not weak, with relatively large randomness. In addition, during the planting stage, the correlation between the Pa and TVDI was the strongest. Changes in the Pa quickly affected the TVDI, with a relatively short lag time. This process was pronounced during the planting stage in 2012. With a deepening meteorological drought from the Pa, the TVDI responded and increased rapidly. Finally, the sensitivity of the drought index to Pa in rain-fed land was higher with a faster response speed, as illustrated in 2010. This is also because rain-fed land has a limited dependence on man-made production, with the main source of water for crop growth being precipitation.

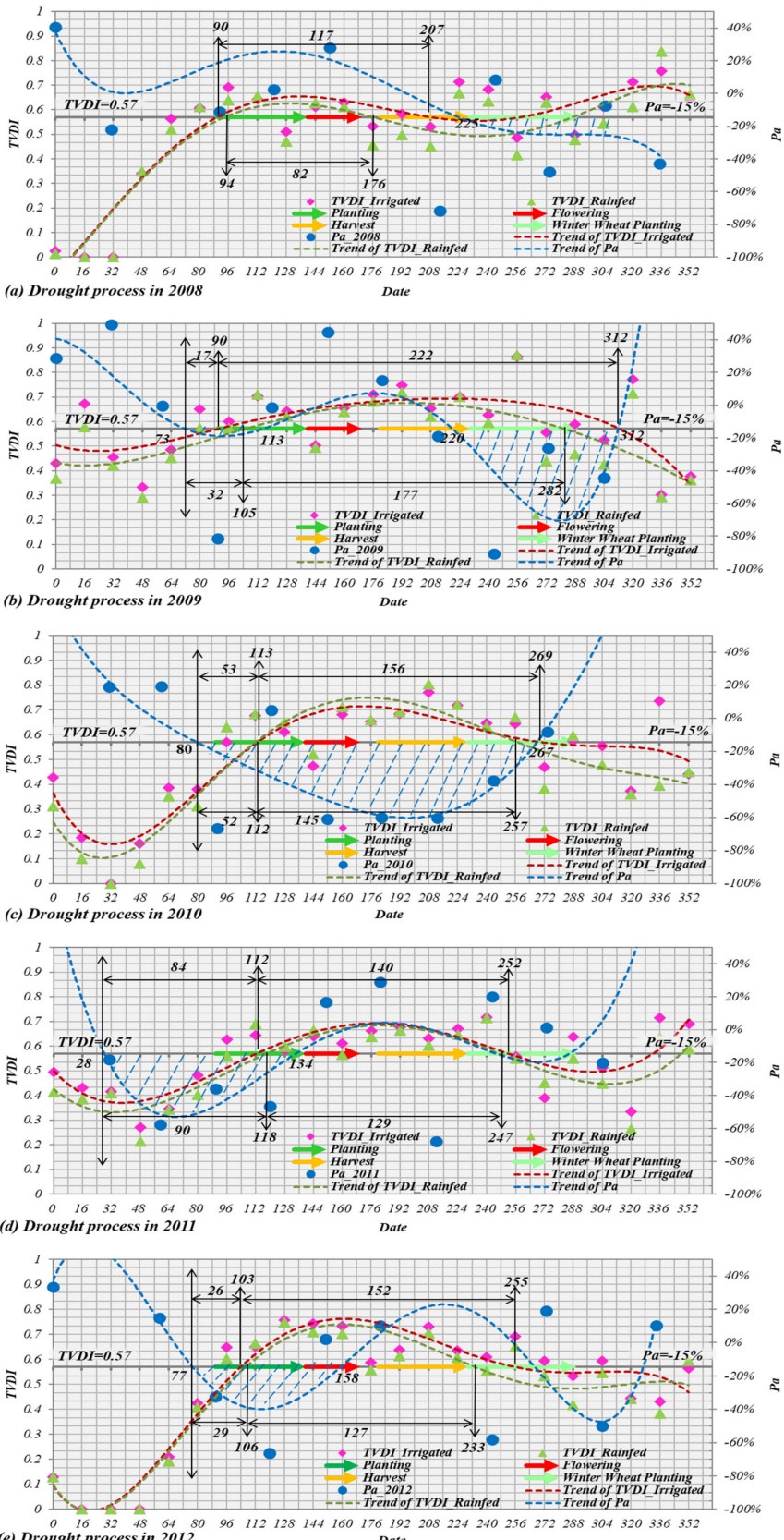

**Figure 8.** Occurrence process and lag relationship between TVDI and precipitation anomaly (Pa). (**a**) Drought process in 2008, (**b**) drought process in 2009, (**c**) drought process in 2010, (**d**) drought process in 2011, and (**e**) drought process in 2012.

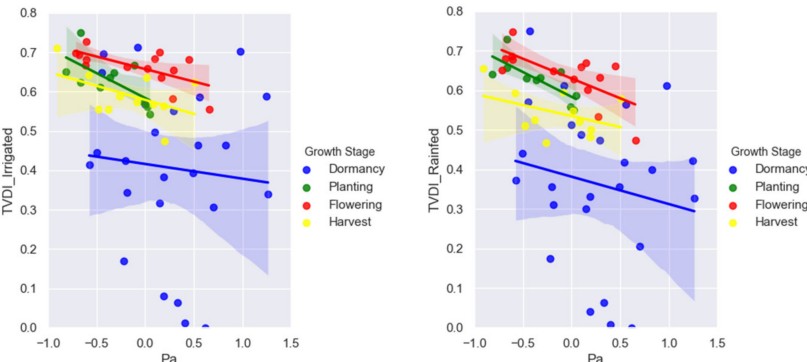

**Figure 9.** Regression fitting for the correlation between Pa and TVDI at different growth stages for 2010 and 2012 data.

### 3.3.2. Forecasting Ability of the Meteorological Drought Index Pa

Figure 10 shows the correlation between the Pa and TVDI over 5 years, with obvious relationships in 2010 and 2012, which indicates that the meteorology and agriculture are closely related in drought years. In wet years, such as 2008 and 2011, the relationship between the Pa and TVDI is relatively weak. However, the time lag as affected by the growth stage and land type allows the meteorological drought to be used as a predictor of agricultural droughts to some extent. The meteorological index uses independent variables constructed from a single rainfall event, which fluctuates greatly throughout the year, so its predictive ability is limited. For example, the meteorological drought in 2009 primarily occurred after the crop harvesting period, while the meteorological drought in 2011 occurred before the sowing period and did not significantly impact crop growth. The meteorological droughts in 2010 and 2012 covered the key growing season for the crops, which affected crop growth. Therefore, it is concluded that the meteorological drought index that covers the crop growth period is more accurate and effective for predicting agricultural droughts.

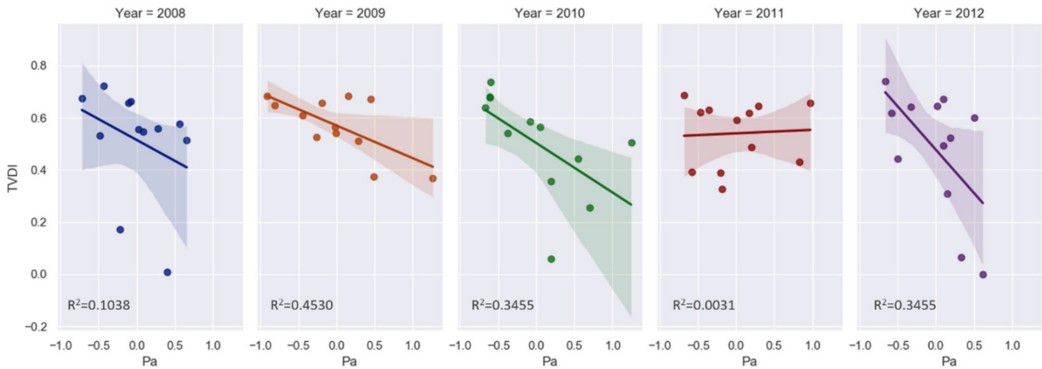

**Figure 10.** Correlation between Pa and TVDI over the 5-year study period.

### 3.4. Analysis of Drought Disasters

The disaster occurrence conditions for three periods of spring from April to May, summer from June to August, and the entire growing season from April to August were compared over the 5-year study period using disaster maps. In addition, the 30-m GlobeLand30-2010 product provided by the National Basic Geographic Information Center was applied to mask the cultivated land and form the 2008–2012 drought disaster zoning map based on cultivated land distributions, as shown in Figure 11.

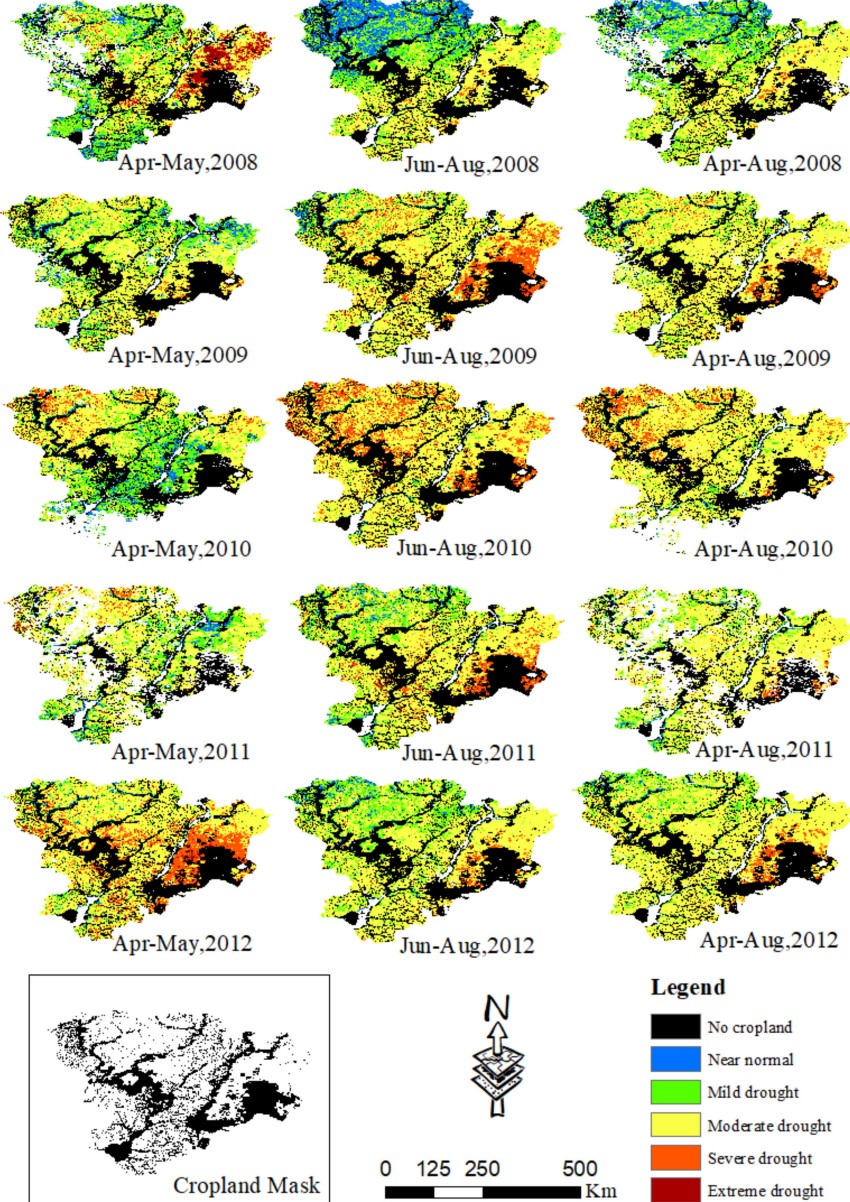

**Figure 11.** The comprehensive drought disaster index (CDDI) for crop growth stages from 2008–2012 in the Volgograd region.

Comparing the CDDI at different periods shows that the drought in this region from 2010 was concentrated in the spring from April to May and summer from June to August. The maximum fire activity was observed during these periods [63,66,69]. The spring drought affects rain-fed dry lands dominated by winter wheat plantations to the north of the river. The spring drought has a certain impact on crop growth as it occurs during crop tilling and jointing. The summer drought affects the entire region, and most crops at this stage enter the key vegetative growth period of heading, booting, and flowering. At this stage, the crop water demand increases, while plant transpiration and soil evaporation are strong. Therefore, a drought during this time has a greater impact on the crop growth and yield. After September, the rainfall increased and the temperature decreased; thus, there was no significant drought over the entire region as the crops were being harvested. The type of drought in the region in 2012 was mainly a spring drought from April to June. The long period of extremely low rainfall in the spring and the rising temperatures continuously reduced the soil moisture. However, the main crops over the entire region were in the initial stage of growth and

required significant amounts of water. Therefore, such hydrothermal conditions have led to drought over large areas. As the precipitation and temperature gradually stabilize after June, the crops return to normal growth conditions.

Due to limitations in the data conditions, only the 2012 open-source disaster data of the Volgograd State drought disaster released by Oxfam [61] were used to statistically analyze the TVDI index drought monitoring results during the key growth period from April to August in 2012 to better understand the monitoring results. Based on the statistical results in Table 5, Volgograd's 9184 km$^2$ arable land experienced a mild drought in 2012, as manifested by near-surface air drying. Then, 59,734 km$^2$ of arable land was in a moderate drought state, as characterized by dry soil surface layers and wilting vegetation leaves. A total of 3571 km$^2$ of arable land was in a state of severe drought, as defined by the presence of thick dry soil layers, obvious wilting, vegetation dryness, and fruit shedding. Finally, 46 km$^2$ of arable land was in a state of extreme drought, as the vegetation appeared significantly dry and dead. Severe and extreme droughts have a greater impact on the crop growth and yield and this is a key area of concern for crop disasters. Such a drought represented 3.33% of the total state area, which is somewhat consistent with the Oxfam result. Oxfam's survey of small-scale farmers in Russia's primary drought-stricken states indicates that the number of farms affected by drought in Volgograd in 2012 was 1584, giving an area of 5,400.072 km$^2$. The main affected items were cereals and livestock, as well as a small number of vegetables, with official estimated losses amounting to 1565.13 million rubles.

**Table 5.** Statistics of areas affected by droughts in the Volgograd region during 2012.

| | Total State Area (km$^2$) | Affected Area from April to August (km$^2$) | Proportion (%) | Levels of Drought Disaster | Judgment Basis |
|---|---|---|---|---|---|
| Statistics from CDDI | 108,448 | 35,913 | 33.12% | Non-arable land or no drought | $0 < CDDI < 2.76$ |
| | | 9184 | 8.47% | Mild drought | $2.76 \leq CDDI < 3.42$ |
| | | 59,734 | 55.08% | Moderate drought | $3.42 \leq CDDI < 4.56$ |
| | | 3571 | 3.29% | Severe drought | $4.56 \leq CDDI < 5.16$ |
| | | 46 | 0.04% | Extreme drought | $5.16 \leq CDDI < 6$ |
| Statistics from Oxfam Report | 113,900 | 5407.042 | 4.75% | none | Farmer survey |

## 4. Discussion

The droughts in Volgograd have been identified and classified from the perspectives of its spatiotemporal evolution patterns, the evolution of rain-fed and irrigation lands, and the lag between meteorological and agricultural droughts. This provides a reference for understanding the overall drought events and guiding agricultural production. However, there are some limitations and deficiencies in this study.

### 4.1. Single Index Selection and Data Limitation

First, only a single agricultural drought index and a single meteorological drought index were used, and there is no relevant comparative discussion and analysis for other types of indices. There are some similarities and differences in the drought monitoring results for different indices. The monitoring effect and lag for different indices require further discussions in future research. Second, the limitations of data indicate that the statistical analysis of disaster situations is only limited to the survey data of farmers from 2012. Although the disaster results are consistent, to a certain extent, the quantitative results are not sufficiently comprehensive. This requires data collection and collation to be strengthened in future research and the real-time processing and accuracy of the data released to be enhanced. In addition, drought events were only studied on a 1-km scale, and different crops were not subdivided. These deficiencies also need to be supplemented and improved in future research.

## 4.2. Gradually Strengthen Lag Analysis

It has been noted that an agricultural drought has a hysteresis to a meteorological drought, and the lag time of each growth period is inconsistent. Specifically, the lag time from heading back to green was the shortest and the lag time of overwintering was the longest. However, the limited amount of data and other factors did not allow the specific delay for each crop growth stage to be calculated. At the same time, the lag time of the drought response in rain-fed and irrigated lands lacks an in-depth analysis. Rain-fed land is more sensitive to the response to a meteorological drought as its water supply is from a single rainfall factor. However, the correlation between rain-fed and irrigated lands was relatively high in Figure 12, and there was no significant lag interval found for the time scale in Figure 13. This result is limited by the temporal resolution of the data and may also be related to the coarse spatial resolution of the rain-fed and irrigated lands. These results should be improved for future research by increasing the data volume or strengthening the time series analysis.

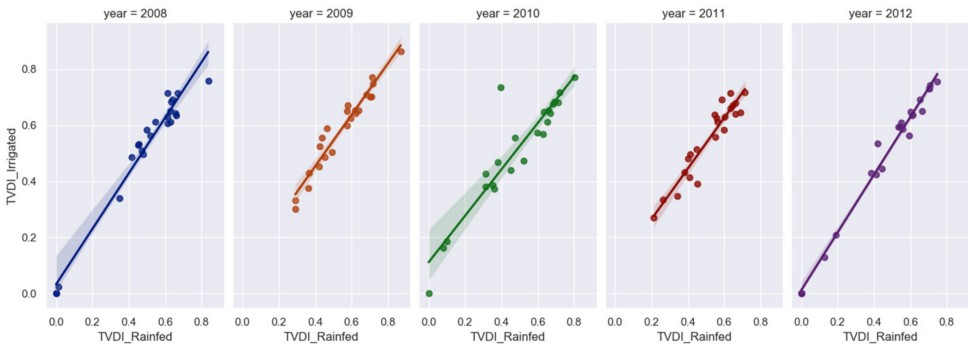

**Figure 12.** Correlation between the TVDIs for the rain-fed and irrigated lands over the five years considered.

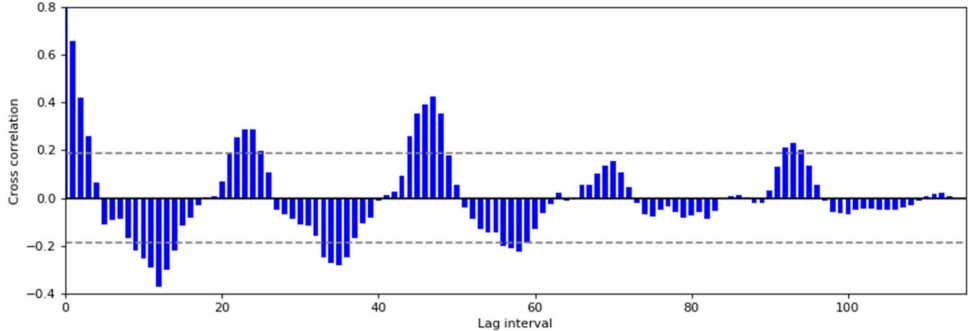

**Figure 13.** Cross correlations between the TVDIs of the rain-fed and irrigated lands over the five years considered.

## 4.3. Supplement and Refinement of Spatial Factors

The characteristics of frequency, intensity, and hysteresis for drought events in real environments may be affected by various spatial factors, such as the topography, landform, soil properties, and crop types. This study only considered two different types of cultivated lands, rain-fed and irrigated, and their local climate. In future research, it will be necessary to refine the scale of spatial factors and analyze the comprehensive response of drought events to multiple factors. In addition, spatial information, such as the DEM or geomorphic type distribution map, can be further combined to analyze the factors that are related to drought characteristics, such as the hysteresis. For example, the time delay for agricultural to meteorological droughts may fluctuate with changes in geographical factors. The distribution of high correlation values between each geographical factor and the agricultural

drought time delay may potentially be related to the factors of poor soil water retention, a high altitude, a steep slope, lower precipitation, and reduced temperatures.

## 5. Conclusions

The monitoring and analysis of agricultural drought events is an important and lasting topic, whether from the perspective of risk analysis or agricultural life guidance. For specific research areas, crop growth will develop with changes in water and heat conditions after droughts, which is reflected in the corresponding variety of parameters for its reflection characteristics. This also provides a basis to use remote sensing drought indices to monitor the spatial evolution of droughts. As the TVDI index can be applied to areas with significant vegetation and temperature changes, it was selected as the remote sensing index for monitoring droughts in this paper. The calculation and visualization of the index allowed a better understanding of the evolution mode of droughts in an area to be developed. Droughts result from changes in the seasons and the spatial development track, which can indicate the drought patterns for an area. Remote sensing drought indices are considered to provide important monitoring capacities for future long-term drought monitoring. The drought monitoring results vary, depending on the different types of crops or the local microclimate. Therefore, the influence of the local microclimate should be discussed.

Meteorological and agricultural droughts describe an event from different perspectives. The meteorological drought index is based on precipitation data with a strong seasonal variability, which illustrates the meteorological conditions of a region. The agricultural drought index describes droughts from the perspective of crop and soil moisture. These differing principles generate distinct time-dependent curves. The rainfall index Pa selected in this paper reflects the general situation of dry conditions in the spring and autumn. When rain is absent, the dry conditions of spring extend into the growing season of summer. In contrast, the TVDI index is more consistent with the growth cycle and temperature level of vegetation, which always shows higher levels in the summer. In general, if there is no precipitation supplement for a period after a meteorological drought, an agricultural drought will follow.

The main growth periods for different crops are not completely synchronous, and the water demand and drought resistance capacities for different growth stages are also variable. From this perspective, studying drought events should not be limited to considering their distributions, but should also focus on whether they constitute disaster events. Droughts can indicate a risk for a given area, while disasters can be used to quantitatively assess the economic loss of the affected area. Therefore, this paper has analyzed both of these aspects to more comprehensively understand the entire drought event.

The above three aspects formed the foci of this study for remote sensing drought monitoring, which utilized the MODIS vegetation index products and surface temperature products to monitor and evaluate droughts in the Volgograd region in Russia by constructing the TVDI and CDDI based on the growth period. Meanwhile, the spatiotemporal differentiation of regional droughts and factors that occur in different types of farming areas with unique growth periods were analyzed and discussed. In addition, a lag analysis was performed for meteorological and agricultural droughts based on the crop growth stages. The main conclusions are as follows.

### 5.1. Identification of Drought Events in 2010 and 2012

The Volgograd region experienced a wide range of drought conditions in 2010 due to abnormal rainfall and high temperatures. The pattern of drought conditions included a spring drought from April to May and a summer drought from June to August. The spring drought first occurred in local areas of the rain-fed drylands, and extreme summer droughts followed due to extreme high temperatures and low precipitation. The degree of continuous droughts during the summer had a stronger impact and wider range as it was during the key growth period of regional crops. The drought in 2012 primarily occurred between April and June and was influenced by extremely dry weather. Such

events significantly impact the growth of the main corn crop in irrigated drylands. As the precipitation and temperature became stable after June, the drought did not experience further strengthening. In addition, the CDDI based on the growth period can correctly indicate the disaster conditions in the region, and the monitoring results are in good agreement with the region's hydro-thermal conditions and related statistics.

### 5.2. Difference in Droughts between Rain-Fed and Irrigation Lands and the Influencing Factors

The drought patterns for surface crops are affected by multiple factors, such as the average surface temperature, extreme climatic factors, and irrigation methods. The research in this paper shows that the temperature is a more important indicator than precipitation for crops, and extreme high-temperature events have a greater negative impact on the crop yield [91]. High temperatures and droughts affect the differences between the crop canopy and the surrounding temperature, which impacts the final crop yield. In normal years, the average temperature level of the irrigated area in the semi-desert region in the southeast was higher than that of the rain-fed land, and the drought index level was higher. Under the extremely high-temperature conditions in 2010, wheat was more sensitive to soil moisture during the germination and filling stages. Therefore, the response of wheat-dominated rain-fed lands to droughts is more obvious, while artificial irrigation regulation in irrigated lands shows lower drought levels in high-temperature climates.

### 5.3. Lag Analysis of Meteorological and Agricultural Droughts

A meteorological drought has a time lag of 1–2 months in comparison to an agricultural drought. Therefore, the meteorological drought index is indicative of drought risk prediction and prevention. However, the correlation between the meteorological drought index Pa and the agricultural drought index TVDI fluctuates greatly during the different crop growth stages, with the strongest correlation during the planting stage and weakest correlation during the dormancy stage. As a result, the meteorological drought in the selected growth period better informs crop drought predictions.

### 5.4. Regional Drought Risk and Irrigation Guidance

From the perspective of drought prevention and risk prediction for local farming, droughts in the southeastern irrigated area are affected by the local climate. This part of the region is more sensitive to precipitation factors, especially during dry conditions, and is less sensitive during the spring. Therefore, the region is prone to drought conditions, which requires stronger irrigation regulations. In contrast, the average surface temperature of the rain-fed land in the northwest is relatively low due to the greater impact of hot summers. Hence, it is necessary to consider the prevention of summer droughts.

In summary, it is possible to effectively monitor the occurrence and spatiotemporal evolution of regional droughts and disasters based on the research presented in this paper. Moreover, a systematic analysis of drought factors and forecast prevention for different types of cultivated land has been achieved, which provides an important theoretical basis and reference for local agricultural production irrigation and scientific management.

**Author Contributions:** Conceptualization, F.C. and H.J.; methodology, Y.H.; software, Y.H.; validation, Y.H. and H.J.; formal analysis, Y.H.; writing—original draft preparation, Y.H.; writing—review and editing, F.C., H.J., L.W., and V.G.B.; visualization, Y.H.; supervision, F.C. and H.J.; project administration, F.C., H.J., and V.G.B.; funding acquisition, F.C., H.J., L.W., and V.G.B. All authors have read and agreed to the published version of the manuscript.

**Funding:** This research was funded by the National Key R&D Program of China, grant number 2017YFE0100800; the International Partnership Program of the Chinese Academy of Sciences, grant number 131211KYSB20170046; and the National Natural Science Foundation of China, grant number 41671505/41871345. The reported study was funded by RFBR, MOST, and DST, according to the research project № 19-55-80021

**Acknowledgments:** The remote sensing data were provided by the National Aeronautics and Space Administration (NASA) and processed in the Google Earth Engine (GEE). The PERSIANN-CDR comes from NOAA National Centers for Environmental Information. The Global Food Security-Support Analysis Data at 30 m (GFSAD30)

was released by the United States Geological Survey (USGS) and the Remote Sensing Mapping Data Products for Global Coverage of 30 m in the 2010 base year was published by the Basic Geographic Information Center.

**Conflicts of Interest:** The authors declare no conflicts of interest.

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
