# Peer review of "Different Drought Legacies of Rain-Fed and Irrigated Croplands in a Typical Russian Agricultural Region"

_remotesensing, doi:10.3390/rs12111700_

Round 1

Reviewer 1 Report

The idea of the paper is great! The submitted research work aims to highlight the role of remote sensing indices in monitoring the drought at regional scales. Several technical aspects of the study were discussed/implemented perfectly and explained sufficiently. The manuscript is clearly structured and the language used is largely appropriate. I see that this manuscript in its form and level can be considered for publication in MDPI-RS after implementing the VERY MINOR COMMENTS that I pointed below.

COMMENTS:

  • The title is adequate for the content of the paper.
  • In the abstract, line 20, correct the word indexes to “INDICES”.
  • Figure 1, try to increase the size of the legends on the bottom left part.
  • Figure 7, text of legends should be increased.
  • Please make sure to define ALL the acronyms form their first appearance in the paper.
  • All references MUST BE CHECKED and formatted as required by MDPI-RS, also make sure that all the references have DOI number unless it is not available.

Author Response

Point 1: In the abstract, line 20, correct the word indexes to “INDICES”.

Response 1: Thanks for pointing this out. Replacements have been made into “indices”. (Line 20, Page1)

Point 2: Figure 1, try to increase the size of the legends on the bottom left part.

Response 2: Thanks for pointing this out. The size of the legends on the bottom left part has been increased for better visualization. (Line 174, Page4)

Point 3: Figure 7, text of legends should be increased.

Response 3: Thanks for pointing this out. The text of the legends has been increased. (Line 393, Page12)

Comments: Please make sure to define ALL the acronyms form their first appearance in the paper.

All references MUST BE CHECKED and formatted as required by MDPI-RS, also make sure that all the references have DOI number unless it is not available.

Response 3: Thanks for pointing this out. All the acronyms and references have been checked to meet journal guideline.

We appreciate for your warm work earnestly, and hope that the correction will meet with approval. Thank you very much for your comments and suggestions.

Reviewer 2 Report

The subject of this paper could have been interesting. However, reading the paper I progressively lost its focus and the aim because sentences, information, and concepts are often confused, random, incomplete, or incorrect. Descriptions are not fluid and are hard to read.

There is an approximation and little attention to the description of indices, parameters, and phenomena, denoting a shallow knowledge of the climatic extreme events and agrometeorology subject, even highlighted by the use of inappropriate terms.

I stop the reading before the end because it became difficult and unnerving. Nevertheless, I give you some recommendations and corrections relative to the section read.

  1. Lines 21-22: change the sentence “the MODIS vegetation index products and surface temperature products…” into “the MODIS vegetation indexes and the surface temperature product…”
  2. Lines 26-27: what do you mean with “sensitivity of meteorological factors to drought”? Drought is a climatic extreme event and generally, drought is influenced by meteorological factors as rainfall and temperature and not the opposite.
  3. Lines 44-46: Probably the definition, causes and impacts of a drought event are not clear. Reformulate the sentence integrating with references.
  4. Lines 47-48: extension (not scopes), duration and severity are characteristics of drought, not characteristics of a trend. You should reformulate the sentence.
  5. Line 51: what are the different “roles” of the four droughts? Do you mean “impacts”?
  6. Line 54: “water shortage” is influenced by prolonged droughts, not the opposite.
  7. Lines 54-68: You must revise the basic concepts of drought and its causes and you should integrate definitions with references. Sentences are redundant and concepts are not clear and not well organized. The main difference between the four types of drought is the duration of the lack/reduction of precipitation and the impacts that progressively are caused. Resume this concept.
  8. Line 70: “constructing meteorological drought indexes…” Delete “meteorological”.
  9. Lines 71-73: Move the sentence “…such as the precipitation anomaly…(SPEI)” after “…drought indexes” in line 70.
  10. Lines 84-85: references are not related to drought indices from remote sensing.
  11. Line 87: you could change the sentence “The remote sensing drought index can be divided as follows” into “Different types of remote sensing drought indices exist. They can be grouped by the type of drought parameters considered (precipitation, moisture, evapotranspiration), the impacts of drought on vegetation (abiotic stresses), or the approach (energy balance, water balance, etc.).
  12. Lines 87-95: Check the grouping. Some indices are not well categorized (see the indices description of the references that you cited). Give some examples and references for group 5. The TVDI is the Temperature Vegetation Dryness (not Drought) Index (see reference 25).
  13. Lines 100-102: What do you mean with “The normalized difference water index (NDWI) can be combined with the shortwave infrared (SWIR) band…to construct the standardized water index (SWI) [35]? SWI is calculated on the basis of NDWI that uses NIR and SWIR in its formula (see reference 35). Reformulate the sentence.
  14. Line 119: This is the first time you mention CDDI in the introduction, without explaining what is it.
  15. Line 125: The coordinates indicate a point, not an area. Please insert in the map 1 the point you described (if represent the geometric center of the region), or delete the coordinates specification.
  16. Lines 128-131: The climatic classification of an area needs to specify not only the mean annual precipitation amount but also the mean annual temperature. Moreover, in line 128 you should describe how is the rainfall distribution during seasons. A map with DEM and river identification could be useful to better focus the region.
  17. Lines 144-156: homogenize the product's description: or you specify the resolution for all the products or you just refer to tables 1 and 2.
  18. Line 200: specify “…the occurrence of drought events is compared…”
  19. Lines 203-204: what do you mean with “regional precipitation deviates from the local average”? An anomaly of a parameter is a deviation from the average in a specific location. Explain better. What is the period that you considered to calculate a meteorological drought (1 month, 2 months, a season,…)?
  20. Sub-paragraph 2.5: the CDDI is probably the newest think of this paper. You should describe in deep what it is and how it is calculated.
  21. Lines 214-216: What is Pi? Is it the precipitation of the i-month or else? When introducing a parameter for the first time you have to describe what is and, eventually, how do you calculate it.
  22. Line 218: What are “the existing crop drought sensitivity settings”? Can you quantify them? How do you obtain Pi weights?
  23. Line 219: the Pi weights that you describe range from April to August. What about September and October (in equation 6 the sum is from April to October)?
  24. Table 4: could you explain the table in the text? No explanation is given on how you calculate the different CDDI thresholds. How can we interpret the title of the second column? The range is from June of a year to the following April? Why May is repeated? Do the differences consist of the length of the period considered?
  25. Line 225: what do you mean with “Crop drought is a concentrated reflection of…”? Crop or Agricultural drought?
  26. Line 226: “…hydrothermal conditions indicated in Figures 4 and 5 over the same period in different years…”. Actually, the two graphs represent different time lengths, the first starting in April and ending in October, the second starting in January and ending in December.
  27. 4: The graph is not clear. For climatic analysis or comparisons between different years, it is better to aggregate daily precipitation values into monthly data and choose other graph typologies.
  28. 5: Even in this case the differences between years are not so evident. It is better to aggregate values at a monthly level.
  29. Line 235: The drought monitoring terminology is not appropriate. Drought monitoring is a quasi-real time/real time monitoring of the occurrence of a drought event. In this case, you are doing a retrospective analysis of a drought occurrence to validate the effectiveness of a methodology/tool/indices/models, in order to use them for operational monitoring.
  30. Line 241: “meteorological and hydrothermal” are redundant.
  31. Lines 243-244: what does it mean “highly correlated from year to year”?
  32. Lines 245-281: the description of phenological stages must be moved to the materials and methods and could be resumed. However, there is a weak knowledge of agrometeorological concepts. Results are only partially and randomly shown.
  33. Fig 7: the graphs are not readable.
  34. Lines 302-303: The sentence is not clear. Explain the concept.
  35. Lines 308-309: How can you affirm that 2010 had “extremely high temperature”? Fig. 5 don’t show extreme differences.
  36. 8: the legends of a), b), d) and e) are incomplete. Use only one legend for all the graphs.
  37. Line 317: “summer torrential drought”? Torrential is referred to a high intense rainfall!

Author Response

Point 1: Lines 21-22: change the sentence “the MODIS vegetation index products and surface temperature products…” into “the MODIS vegetation indexes and the surface temperature product…”

Response 1: Thank you for pointing this out. The sentences have been changed to meet your comments. (Line 22, Page1)

Point 2: Lines 26-27: what do you mean with “sensitivity of meteorological factors to drought”? Drought is a climatic extreme event and generally, drought is influenced by meteorological factors as rainfall and temperature and not the opposite.

Response 2: Thank for pointing this out and I agree with your comments. There is a writing mistake and has been rectified to meet your comments. (Line 26, Page1)

Point 3: Lines 44-46: Probably the definition, causes and impacts of a drought event are not clear. Reformulate the sentence integrating with references.

Response 3: Thank for pointing this out. The sentences have been reformulated and integrated with relative references (Richard&Heim,2002). (Line 44-46, Page2)

Point 4: Lines 47-48: extension (not scopes), duration and severity are characteristics of drought, not characteristics of a trend. You should reformulate the sentence.

Response 4: Thank you for pointing this out. There is a writing mistake and the sentence has been reformulated following your comment. (Line 51, Page2)

Point 5: Line 51: what are the different “roles” of the four droughts? Do you mean “impacts”?

Response 5: Thank you for pointing this out. The purpose of this sentence is to illustrate that different types of drought occur at different stages of the water cycle, thus causing different spatiotemporal effects. The word of the sentence has been revised to “impacts”. (Line 54, Page2)

Point 6: Line 54: “water shortage” is influenced by prolonged droughts, not the opposite.

Response 6: Thank you for pointing out this mistake. The expression of the sentence has been modified. (Line 56, Page2)

Point 7: Lines 54-68: You must revise the basic concepts of drought and its causes and you should integrate definitions with references. Sentences are redundant and concepts are not clear and not well organized. The main difference between the four types of drought is the duration of the lack/reduction of precipitation and the impacts that progressively are caused. Resume this concept.

Response 7: Thanks a lot for your criticism and comments. The basic concepts of drought and its causes have been revised. We also integrated definitions and amended statements according to references. Hoping to meet your requirements. (Line 58-85, Page2)

Point 8: Line 70: “constructing meteorological drought indexes…” Delete “meteorological”.

Response 8: Thank you for pointing this out. We have deleted this word under your suggestion. (Line 87, Page2)

Point 9: Lines 71-73: Move the sentence “…such as the precipitation anomaly…(SPEI)” after “…drought indexes” in line 70.

Response 9: Thank you for pointing this out. We have modified this sentence under your suggestion. (Line 87-89, Page2)

Point 10: Lines 84-85: references are not related to drought indices from remote sensing.

Response 10: Thank you for pointing this out. This part refers to the literature on heat waves and forest fires, which does not seem to be strongly related to drought monitoring. The real purpose, however, is to provide more information and open thinking about RS techniques for monitoring arid areas and extremely hot surface pixels. This is also the advantage of RS technology to provide more valuable surface pixel information compared with the meteorological index above. Hope this answer can reach your approval. (Line 103-105, Page3)

Point 11: Line 87: you could change the sentence “The remote sensing drought index can be divided as follows” into “Different types of remote sensing drought indices exist. They can be grouped by the type of drought parameters considered (precipitation, moisture, evapotranspiration), the impacts of drought on vegetation (abiotic stresses), or the approach (energy balance, water balance, etc.).”

Response 11: Thank you for pointing this out. We have modified this sentence under your suggestion. (Line 109-112, Page3)

Point 12: Lines 87-95: Check the grouping. Some indices are not well categorized (see the indices description of the references that you cited). Give some examples and references for group 5. The TVDI is the Temperature Vegetation Dryness (not Drought) Index (see reference 25).

Response 12: Thank you for pointing this out. The group of indices has been modified and supplemented with references. Examples and references were added in group 5. Hope this answer can reach your approval. (Line 112-128, Page3)

Point 13: Lines 100-102: What do you mean with “The normalized difference water index (NDWI) can be combined with the shortwave infrared (SWIR) band…to construct the standardized water index (SWI) [35]”? SWI is calculated on the basis of NDWI that uses NIR and SWIR in its formula (see reference 35). Reformulate the sentence.

Response 13: Thank you for pointing this out. The original sentence was incorrectly expressed and has been modified. (Line 134, Page3)

Point 14: Line 119: This is the first time you mention CDDI in the introduction, without explaining what is it.

Response 14: Thank you for pointing this out. The complete spelling has been added. (Line 152, Page4)

Point 15: Line 125: The coordinates indicate a point, not an area. Please insert in the map 1 the point you described (if represent the geometric center of the region), or delete the coordinates specification.

Response 15: Thank you for pointing this out. The coordinates have been deleted. (Line 158, Page4)

Point 16: Lines 128-131: The climatic classification of an area needs to specify not only the mean annual precipitation amount but also the mean annual temperature. Moreover, in line 128 you should describe how is the rainfall distribution during seasons. A map with DEM and river identification could be useful to better focus the region.

Response 16: Thank you for pointing this out. The mean annual temperature has been added and rainfall distribution during seasons has been described. A map with DEM and river identification has been supplemented to better focus the region. (Line 162,165,174, Page4)

Point 17: Lines 144-156: homogenize the product's description: or you specify the resolution for all the products or you just refer to tables 1 and 2.

Response 17: Thanks for pointing this out. We have homogenized the product's description. (Line 194,200,202, Page5)

Point 18: Line 200: specify “…the occurrence of drought events is compared…”

Response 18: Thanks for pointing this out. The missing word has been added. (Line 247, Page7)

Point 19: Lines 203-204: what do you mean with “…regional precipitation deviates from the local average”? An anomaly of a parameter is a deviation from the average in a specific location. Explain better. What is the period that you considered to calculate a meteorological drought (1 month, 2 months, a season,…)?

Response 19: Thanks for pointing this out. The sentence has been modified and every month Pa was calculated. (Line 250, Page7)

Point 20: Sub-paragraph 2.5: the CDDI is probably the newest think of this paper. You should describe in deep what it is and how it is calculated.

Response 20: Thanks for pointing this out. Relevant information has been supplemented about CDDI. (Line 250, Page7)

Point 21: Lines 214-216: What is Pi? Is it the precipitation of the i-month or else? When introducing a parameter for the first time you have to describe what is and, eventually, how do you calculate it.

Response 21: Thanks for pointing this out. Pi is the weight value of each month, which was set with relevant reference [80]. The research area in paper [80] is similar to that in this paper in latitude, climate, crop type and phenological calendar, so the weight setting is referred to and adjusted to this paper. (Line 268-270, Page7)

Point 22: Line 218: What are “the existing crop drought sensitivity settings”? Can you quantify them? How do you obtain Pi weights?

Response 22: Thanks for pointing this out. The existing crop drought sensitivity settings was referred to Gon’s literature [80]. According to Gon’s article, if a crop is affected by a lack of rain only at a certain stage of growth, the impact is usually limited, and the loss of grain yield is usually no more than 30% to 40%. Therefore, by comparing the correlation between the lack of rain and the grain yield, Gon defined the lack of rain index as 1 for the growth stage causing more than 30% of the grain loss, and 0.3 for the growth stage causing 10% of the grain loss, and then gave the weight of each growth stage of different crops. On this basis, this paper adjusted and set the Pi value in combination with the crop type and growing season in the research area. (Line 268, Page7)

Point 23: Line 219: the Pi weights that you describe range from April to August. What about September and October (in equation 6 the sum is from April to October)?

Response 23: Thanks for pointing this out. The equation 6 has been rectified. There are 3 reasons to choose the period from April to August. First of all, the main growth season of crops in Volgograd state is from April to August, and most of the crops are harvested after entering September. Second, lower temperatures after September greatly reduce the risk of drought disaster for crops. Finally, CDDI for September and October was calculated during the mapping process of the earlier article. However, due to the image quality in the western region (such as the missing image in the northwest in 2010 and the poor image synthesis quality in the southwest in 2012, seen in figure 6), the mapping effect is not ideal. Therefore, this paper selected April to August as the time range for CDDI index calculation. (Line 267, Page7)

Point 24: Table 4: could you explain the table in the text? No explanation is given on how you calculate the different CDDI thresholds. How can we interpret the title of the second column? The range is from June of a year to the following April? Why May is repeated? Do the differences consist of the length of the period considered?

Response 24: Thanks for pointing this out. There is a writing mistake of title of the second column and we have rectified it into right version. Table 4 shows the grading criteria used for CDDI visualization. The first three columns represent the threshold of CDDI at different time periods, in order to compare the comprehensive disaster condition at different growth stages. For example, the third column represents the calculation of CDDI values over the entire growth season during April to August. The threshold of CDDI level is obtained by linear weighting the upper and lower limits of the threshold interval of TVDI in table 3 with the weight Pi. (Line 272, Page8)

Point 25: Line 225: what do you mean with “Crop drought is a concentrated reflection of…”? Crop or Agricultural drought?

Response 25: Thanks for pointing this out. What this sentence wants to express is that crop drought is a response to the hydrothermal conditions in a region during a period of time. If the crops are not in drought, then the local hydrothermal conditions are relatively good during this period. (Line 275, Page8)

Point 26: Line 226: “…hydrothermal conditions indicated in Figures 4 and 5 over the same period in different years…”. Actually, the two graphs represent different time lengths, the first starting in April and ending in October, the second starting in January and ending in December.

Response 26: Thanks for pointing this out. The time lengths of these two graphs have been rectified. (Line 276, Page8)

Point 27: 4: The graph is not clear. For climatic analysis or comparisons between different years, it is better to aggregate daily precipitation values into monthly data and choose other graph typologies.

Response 27: Thanks for pointing this out. The original figure 4 has been modified to a bar chart of monthly cumulative precipitation, so as to better reflect the inter-annual comparison. (Line 288, Page8)

Point 28: 5: Even in this case the differences between years are not so evident. It is better to aggregate values at a monthly level.

Response 28: Thanks for pointing this out. The original figure 5 has been modified to a bar chart of monthly average temperature, so as to better display temperature anomaly in 2010 and 2012. (Line 291, Page8)

Point 29: Line 235: The drought monitoring terminology is not appropriate. Drought monitoring is a quasi-real time/real time monitoring of the occurrence of a drought event. In this case, you are doing a retrospective analysis of a drought occurrence to validate the effectiveness of a methodology/tool/indices/models, in order to use them for operational monitoring.

Response 29: Thanks for pointing this out. The expression of sentences has been adjusted to convey information more accurately and rationally. (Line 285, Page8)

Point 30: Line 241: “meteorological and hydrothermal” are redundant.

Response 30: Thanks for pointing this out. The redundant words have been deleted under your suggestion. (Line 294, Page9)

Point 31: Lines 243-244: what does it mean “highly correlated from year to year”?

Response 31: Thank you for pointing this out. This sentence was incorrectly expressed and has been deleted. (Line 296, Page9)

Point 32: Lines 245-281: the description of phenological stages must be moved to the materials and methods and could be resumed. However, there is a weak knowledge of agrometeorological concepts. Results are only partially and randomly shown.

Response 32: Thanks for pointing this out. The description of phenological stages has been moved in paragraph 2.1 under your suggestion. Knowledge of agrometeorological concepts have been supplemented and the expression of the sentence is logically enhanced from the three elements of climate, soil and vegetation. Hope this answer can reach your approval. (Line 298-344, Page9 and Line 175-187, Page5)

Point 33: Fig 7: the graphs are not readable.

Response 33: Thanks for pointing this out. Each sub-diagram in figure 7 shows the time series of TVDI and LST of irrigated land and rainfed land in that year. TVDI value was marked with different colors according to the classification criteria, so as to visually show the intensity and evolution of the drought condition. According to the chart, it can be found that the time evolution pattern and LST level of TVDI in irrigated land and rainfed land are different, thus assisting the subsequent analysis. (Line 393, Page11)

Point 34: Lines 302-303: The sentence is not clear. Explain the concept.

Response 34: Thanks for pointing this out. This statement is intended to explain that the occurrence of drought events is influenced by the climate factors.

On the one hand, the influencing factor comes from the geographical location and the overall climate condition of the region. For example, in this paper, the irrigated land is located nearby the southern desert with a considerable high level of average temperature, so the high temperature stress on crops in irrigated land is relatively stronger than that in rainfed land.

On the other hand, extreme weather events are also important factors in inducing drought. For example, the extreme lack of rain in 2010 and the extreme high temperature in these two years both caused severe moisture and temperature stress to crop growth. (Line 373, Page11)

Point 35: Lines 308-309: How can you affirm that 2010 had “extremely high temperature”? Fig. 5 don’t show extreme differences.

Response 35: Thanks for pointing this out. State of the climate in 2010(Blunden, J.,2011, pp200) points out the seasonal anomalies and exceptionally extensive heat wave during summer (June to August, JJA) in Eastern Europe. The word has been changed to “exceptionally” for a more reasonable statement. (Line 378, Page11)

Point 36: 8: the legends of a), b), d) and e) are incomplete. Use only one legend for all the graphs.

Response 36: Thanks for pointing this out this mistake. The legend for all graphs has been modified to be complete and consistent. (Line 427, Page12)

Point 37: Line 317: “summer torrential drought”? Torrential is referred to a high intense rainfall!

Response 37: Thanks for pointing this out this mistake. The incorrect word has been deleted. (Line 387, Page11)

We appreciate for your warm work earnestly, and hope that the correction will meet with approval. Thank you very much for your comments and suggestions.

Reviewer 3 Report

The manuscript has presented a case study over a Russian agricultural region on meteorological and agricultural drought. It attempted to evaluate droughts using two existing drought indices, namely, TVDI and precipitation anomaly (Pa). The drought phenomena was computed using above indices are generally widely used. The authors presented a well-defined methods for drought assessment. However, it has add some new dimensions, such as, evaluation between rain-fed and irrigated conditions, time-lag between meteorological and agricultural droughts, TVDI and precipitation anomaly at different growth stages. Results are well explained and supported by data selection. However, I have added some concerns about the manuscript which can be useful for authors to improve manuscript. 

Comments:

  1. In Abstract: It is mostly subjective and quantitative evidences of main findings are missing?
  2. In section 2.2, data were having spatial resolution that varied 1 km to 0.25 degree. There is no clarity in the manuscript about final resolution of VTCI and Pa as both are compared for establishing relationships between TVDI and precipitation anomaly at different growth stages.
  3. In section 2.3 and TVDI equations 1, some of the variables are not described like Tsmin and Tmin. Is it LST or air temperature?
  4. There was standard TVDI index provided and it has been widely used. Whether your method is a modified one, then it should mention like mTVDI. In case, the final eq. no. 4 is same with the existing one, then this section should be shortened. In eq. no. 4, again Tmin is not defined.  
  1. In Equation 6 (CDDI), weights Pi has been given as 0.4, 0.5, 0.8, 0.9, and 0.4 from April to August. What is the logic of these values? Can it be used in other sites or countries? Moreover, 5 months was used for CDDI calculation and I-value should vary from 4 to 8. But it was mention as 4 to 10.
  1. Table 4: How these values are applicable in other regions. I think this will vary with selected Pi.  
  2. Figure 6: TVDI inversion means 1.0 will indicate wet edge? I think it is simple TVDI.
  3. Figure 7: Its complicated to understand because of many variables are fragmented. Is there any way for simplification? There are two R2 but which one is rain-fed and irrigated. Moreover, in the caption you should write over the years 2008 to 2012.
  4. In Figure 7, TVDI threshold is 0.46 and further changed to 0.57 in Figure 8. It is not clear why threshold has been changed?
  5. Section 3.3.2: R2 for each years may be given.
  6. In Figure 9 caption, write at different growth stages of 2010 and 2012 data was shown.
  7. In Figure 10: How these points (12 samples) are created?

Author Response

Point 1: In Abstract: It is mostly subjective and quantitative evidences of main findings are missing?

Response 1: Thank you for pointing this out. Main conclusions of different sections are presented in the abstract, so that the research content and thinking of this paper can be quickly understand. The detailed chart analysis and quantitative description are described in the corresponding sections, so the detailed data are not presented in the abstract. I hope this answer will satisfy your question. (Line 29, Page1)

Point 2: In section 2.2, data were having spatial resolution that varied 1 km to 0.25 degree. There is no clarity in the manuscript about final resolution of VTCI and Pa as both are compared for establishing relationships between TVDI and precipitation anomaly at different growth stages.

Response 2: Thank you for pointing this out. TVDI is calculated based on NDVI and LST, as a result, the resolution of the calculated TVDI is consistent with 1km. The mean value is adopted to discuss the relationship between TVDI and Pa on a time scale, thus spatial resolution is not considered in this section. (Line 190, Page5)

Point 3: In section 2.3 and TVDI equations 1, some of the variables are not described like Tsmin and Tmin. Is it LST or air temperature?

Response 3: Thanks for pointing this out. There is a writing mistake and has been rectified. (Line 215, Page6)

Point 4: There was standard TVDI index provided and it has been widely used. Whether your method is a modified one, then it should mention like mTVDI. In case, the final eq. no. 4 is same with the existing one, then this section should be shortened. In eq. no. 4, again Tmin is not defined

Response 4: Thanks for pointing this out. There is a redundant description and has been deleted. (Line 228-231, Page6)

Point 5: In Equation 6 (CDDI), weights Pi has been given as 0.4, 0.5, 0.8, 0.9, and 0.4 from April to August. What is the logic of these values? Can it be used in other sites or countries? Moreover, 5 months was used for CDDI calculation and I-value should vary from 4 to 8. But it was mention as 4 to 10.

Response 5: Thanks for pointing this out. The existing crop drought sensitivity settings was referred to Gon’s literature [80]. According to Gon’s article, if a crop is affected by a lack of rain only at a certain stage of growth, the impact is usually limited, and the loss of grain yield is usually no more than 30% to 40%. Therefore, by comparing the correlation between the lack of rain and the grain yield, Gon defined the lack of rain index as 1 for the growth stage causing more than 30% of the grain loss, and 0.3 for the growth stage causing 10% of the grain loss, and then gave the weight of each growth stage of different crops.

The setting of Pi should take into account the local crop type and phenological calendar, so it needs to be adjusted according to the actual situation in different sites or countries. The research area in paper [80] is similar to that in this paper in latitude, climate, crop type and phenological calendar, so the weight setting is referred to and adjusted to this paper.

The equation 6 has been rectified. There are 3 reasons to choose the period from April to August. First of all, the main growth season of crops in Volgograd state is from April to August, and most of the crops are harvested after entering September. Second, lower temperatures after September greatly reduce the risk of drought disaster for crops. Finally, CDDI for September and October was calculated during the mapping process of the earlier article. However, due to the image quality in the western region (such as the missing image in the northwest in 2010 and the poor image synthesis quality in the southwest in 2012, seen in figure 6), the mapping effect is not ideal. Therefore, this paper selected April to August as the time range for CDDI index calculation. (Line 267, Page7)

Point 6: Table 4: How these values are applicable in other regions. I think this will vary with selected Pi.

Response 6: Thanks for pointing this out. For areas where crop types and phenological calendars are relatively close, the setting of values can be adjusted based on the relevant references and existing grading standards. For regions with different climate types and crop planting patterns, sensitivity factors can be determined according to their response of crop yield or physiological morphological indicators to climate factors. (Line 272, Page8)

Point 7: Figure 6: TVDI inversion means 1.0 will indicate wet edge? I think it is simple TVDI.

Response 7: Thanks for pointing this out. The TVDI value varies from 0–1. The closer the value of TVDI was to 1, the more severe the drought was. The different colors in legend correspond to the TVDI levels in table 3. (Line 352, Page10)

Point 8: Figure 7: Its complicated to understand because of many variables are fragmented. Is there any way for simplification? There are two R2 but which one is rain-fed and irrigated. Moreover, in the caption you should write over the years 2008 to 2012.

Response 8: Thanks for pointing this out. Each sub-diagram in figure 7 shows the time series of TVDI and LST of irrigated land and rainfed land in that year. TVDI value was marked with different colors according to the classification criteria, so as to visually show the intensity and evolution of the drought condition. According to the chart, it can be found that the time evolution pattern and LST level of TVDI in irrigated land and rainfed land are different, thus assisting the subsequent analysis. Moreover, R2 has been labeled and the title has been supplemented in the caption. (Line 393, Page12)

Point 9: In Figure 7, TVDI threshold is 0.46 and further changed to 0.57 in Figure 8. It is not clear why threshold has been changed?

Response 9: Thanks for pointing this out. The distribution of TVDI> 0.57 in Figure 7 is the most consistent with the growing season range in the crop phenology. For 0.46 <TVDI <0.57, some points appeared before the start of the growing season. In order to determine the onset time of regional drought more accurately and avoid premature estimation of the drought situation, the threshold of 0.57 is selected in Figure 8 as the beginning threshold of the true drought on the surface. (Line 397, Page12)

Point 10: Section 3.3.2: R2 for each year may be given.

Response 10: Thanks for pointing this out. R2 for each year has been supplemented into Figure 10 under your suggestion. (Line 450, Page14)

Point 11: In Figure 9 caption, write at different growth stages of 2010 and 2012 data was shown.

Response 11: Thanks for pointing this out. The caption has been supplemented under your suggestion. (Line 448, Page14)

Point 12: In Figure 10: How these points (12 samples) are created?

Response 12: Thanks for pointing this out. These points represent Pa and mean value of TVDI for each month. (Line 464, Page14)

We appreciate for your warm work earnestly, and hope that the correction will meet with approval. Thank you very much for your comments and suggestions.

Round 2

Reviewer 2 Report

The authors have answered adequately to all my comments.

Author Response

Dear Reviewer,

We appreciate for your warm work earnestly. Thank you very much for your comments.

Kind regards,
Ms. He
E-Mail: [email protected]

Reviewer 3 Report

The authors have improved the manuscript and some minor correction is required.  

1) L215 and Eq. 1 needs citation for TVDI

2) Table 4: June to May, verify it. It could be May to June?

3) Figure 6: TVDI inversion in caption:  Here Inversion is not right, Inversion means 0 becomes 1 and 1 becomes 0. Claryify this. 

Author Response

Point 1: 1) L215 and Eq. 1 needs citation for TVD

Response 1: Thanks for pointing this out. A reference has been added under your suggestion. (Line 215, Page6)

Point 2: 2) Table 4: June to May, verify it. It could be May to June?

Response 2: Thanks for pointing this out. There is a writing mistake and has been revised. (Line 272, Page8)

Point 3: 3) Figure 6: TVDI inversion in caption: Here Inversion is not right, Inversion means 0 becomes 1 and 1 becomes 0. Claryify this.

Response 3: Thanks for pointing this out. The caption in figure 6 has been modified. Hope this answer can reach your approval. (Line 352, Page11)

We appreciate for your warm work earnestly, and hope that the correction will meet with approval. Thank you very much for your comments and suggestions.
